# Assessment of Serum Endocan Levels and Their Associations with Arterial Stiffness Parameters in Young Patients with Systemic Lupus Erythematosus

**DOI:** 10.3390/jcm14175955

**Published:** 2025-08-23

**Authors:** Ágnes Diószegi, Hajnalka Lőrincz, Eszter Kaáli, Sára Csiha, Judit Kaluha, Éva Varga, Dénes Páll, Tünde Tarr, Mariann Harangi

**Affiliations:** 1Division of Metabolic Diseases, Department of Internal Medicine, University of Debrecen Faculty of Medicine, 4032 Debrecen, Hungary; 2Department of Medicine, Västerviks Sjukhus Hospital, 593 33 Västerviks, Sweden; 3Department of Internal Medicine and Hematology, Semmelweis University, 1083 Budapest, Hungary; 4Department of Medical Clinical Pharmacology, University of Debrecen, 4032 Debrecen, Hungary; 5Division of Clinical Immunology, Department of Internal Medicine, University of Debrecen Faculty of Medicine, 4032 Debrecen, Hungary; 6ELKH-UD Vascular Pathophysiology Research Group 11003, University of Debrecen, 4032 Debrecen, Hungary; 7Institute of Health Studies, Faculty of Health Sciences, University of Debrecen, 4032 Debrecen, Hungary

**Keywords:** endocan, tumor necrosis factor-alpha, C-reactive protein, atherosclerosis, systemic lupus erythematosus, arterial stiffness, pulse wave velocity

## Abstract

**Background**: Systemic lupus erythematosus (SLE) is an autoimmune disorder associated with premature atherosclerosis and vascular impairment. However, the role of endocan, a biomarker of glycocalyx injury, is not completely clarified in the detection of vascular damage. Therefore, our aim was to investigate serum endocan in comparison with conventional inflammatory markers, arterial stiffness parameters, and carotid ultrasound findings in a cohort of young patients with SLE. **Methods**: We enrolled 47 clinically active young SLE patients (40 females and 7 males) in the study. Arterial stiffness indicated by augmentation index and pulse wave velocity (PWV) was measured by arteriography. Brachial artery flow-mediated dilatation and common carotid intima-media thickness were detected by ultrasonography. The serum concentrations of endocan, IL-6, MPO, MCP-1, MMP-3, -7, and -9, as well as TNFα, were measured by an enzyme-linked immunosorbent assay (ELISA). **Results**: We found significant negative correlations between serum endocan and both CH50 and C3. Serum endocan was higher in active SLE patients compared to inactive patients, however, the difference was not statistically significant (241.4 (183–295) vs. 200.3 (167–278) pg/mL; *p* = 0.313). Serum TNFα and hsCRP significantly correlated with PWV. However, we did not detect significant correlations between vascular diagnostic tests and serum endocan levels. **Conclusions**: Based on our results, serum endocan is associated with disease activity; however, further studies are needed to clarify the value of serum endocan in the cardiovascular risk estimation of SLE patients. Measurement of serum endocan, as well as the routine assessment of arterial stiffness parameters, should be integrated into the comprehensive management plans of young patients with SLE.

## 1. Introduction

A global cause of mortality, cardiovascular disease (CVD) is particularly prevalent in cases of systemic lupus erythematosus (SLE). Individuals with SLE exhibit an all-cause mortality rate approximately 2.6 times greater than that observed in the general population, with CVD contributing to a standardized mortality ratio exceeding 2 [1]. Despite this elevated risk, commonly implemented CVD risk prediction models—such as the Framingham Risk Score and the American Heart Association’s ASCVD calculator—consistently underestimate cardiovascular risk in adult SLE patients, particularly in women. Accurate risk stratification is even more complex in pediatric and young adult populations with juvenile-onset SLE [2,3].

Indeed, beside traditional cardiovascular risk factors including hypertension, obesity, diabetes, smoking, age, and male gender, enhanced atherogenesis in SLE may be facilitated by non-traditional risk elements, such as systemic inflammation, antiphospholipid antibodies, renal involvement, and characteristic “lupus pattern” dyslipidemia, as well as medical treatment including corticosteroids [4,5,6]. The clinical significance and prognostic role of novel vascular, metabolic, and inflammatory biomarkers in individuals with SLE are under investigation. A recently published study demonstrated that concentrations of glycine, omega-3/6 ratio, medium-density LDL cholesterol, and intermediate-density lipoprotein-cholesterol were strong metabolite features associated with plaque status, characterized by carotid intima-media thickness (cIMT) progression in adult and juvenile SLE [7].

In recent years, the role of endothelial glycocalyx integrity in the development of cardiovascular diseases has received increasing attention. The endothelial glycocalyx is a negatively charged, carbohydrate-rich matrix located on the luminal surface of endothelial cells, functioning as a semi-permeable barrier that modulates molecular and cellular interactions between circulating blood components and the vascular endothelium [8]. It is a complex, filamentous coating providing scaffolding for the underlying endothelial cell, and composed of proteoglycans, glycoproteins, and glycosaminoglycans [9]. Dysfunction or disruption of the endothelial glycocalyx in various pathological conditions leads to impaired endothelial function and contributes to the development of cardiovascular diseases [8]. Endocan, or endothelial cell-specific molecule-1 (ESM-1), is a soluble dermatan sulfate proteoglycan predominantly released by activated endothelial cells and is also expressed in the endothelium of organs such as the lungs and kidneys [10]. Unlike many other dermatan sulphate proteoglycans, endocan has low molecular weight (approximately 50 kDa), is soluble (not linked to the cell membrane), and thus detectable in circulation [11]. The synthesis and secretion of endocan are upregulated by pro-inflammatory cytokines, such as tumor necrosis factor-alpha (TNFα) [12]. It has been shown that endocan can upregulate the expression of other key molecules by inducing the nuclear factor-κB (NF-κB) signaling pathway, including matrix metalloproteinase (MMP)-9 and matrix metalloproteinase-13 [13], while the expression of matrix metalloproteinase-1 and -2 were not affected [14].

Endocan has recently emerged as a novel biomarker indicative of endothelial cell activation and is implicated in the regulation of leukocyte adhesion to the vascular endothelium. Previous studies have reported that in patients with cardiovascular disease, endocan levels are significantly increased [15,16]. The possible role of endocan in the pathogenesis of SLE-associated vascular complications is not fully clarified. Increased serum endocan was detected in SLE patients compared to healthy volunteers. Furthermore, the authors reported a positive correlation between serum endocan and interleukin-1 beta [17]. Serum endocan was higher in SLE patients than in controls and positively correlated with cIMT in women with SLE [18]. However, a comprehensive assessment involving serum endocan measurement, evaluation of a wide spectrum of inflammatory and vascular markers, and the detection of early vascular damage in SLE patients has not yet been conducted. Furthermore, to date, the correlation between arterial stiffness parameters, brachial artery flow-mediated dilatation (FMD), and serum endocan has not been investigated in patients with SLE.

Since inaccurate risk assessment is especially prevalent in young SLE patients, we aimed to evaluate the association between the glycocalyx injury biomarker endocan as well as traditional inflammatory markers and early vascular damage detected by arteriography, FMD, and carotid artery ultrasonography in young SLE patients. We hypothesized that serum endocan is associated with early vascular damage and would provide a better predictor compared to widely used inflammatory parameters, including high-sensitivity C-reactive protein (hsCRP) and TNFα.

## 2. Materials and Methods

### 2.1. Patients

A total of 47 patients with clinically active systemic lupus erythematosus (SLE) were included in the study, comprising 40 women and 7 men, all receiving care at the Division of Clinical Immunology, Department of Internal Medicine, University of Debrecen. All participants met the 2019 EULAR/ACR classification criteria for SLE [19]. Disease activity was assessed using the SLE Disease Activity Index (SLE DAI), with scores categorized as follows: 0 indicating inactive disease, 1–5 mild activity, 6–10 moderate activity, 11–19 high activity, and ≥20 representing very high disease activity. In the statistical analysis, SLE DAI scores ≥ 6 were considered “active,” while an SLE DAI score < 6 determined the “inactive” patient group. We excluded patients with active lupus nephritis, acute bacterial, viral or fungal infections, severe comorbidities, pregnancy, lactation, and malignant disease. The included patients had no previous major cardiovascular events, including acute myocardial infarction, significant stenosis of the carotid artery, or ischemic stroke. Appendix A provides a summary of the patients’ clinical profiles and associated comorbid conditions.

Informed written consent was obtained from all participants prior to inclusion in the study. The central laboratory received authorization from the National Public Health and Medical Officer Service (approval no. 094025024). Ethical approval was granted by the Regional Ethics Committee of the University of Debrecen (reference: DE RKEB/IKEB 4775-2017; approval date: 3 April 2020) and by the Hungarian Medical Research Council (ETT TUKEB 34952-1/2017/EKU; approval date: 30 June 2017). Participant recruitment was conducted between 5 September 2020 and 10 December 2022.

### 2.2. Collection of Blood Samples and Standard Laboratory Analyses

Venous blood samples were obtained after a 12-h overnight fast. Laboratory markers—including high-sensitivity C-reactive protein (hsCRP), serum amyloid A, total cholesterol, triglycerides, HDL-C, LDL-C, apolipoprotein A1, and apolipoprotein B100—were measured from fresh serum using Cobas c600 analytical systems (Roche Ltd., Mannheim, Germany). Autoantibodies such as anti-double-stranded DNA (dsDNA), anti-beta-2-glycoprotein I (B2GPI), and anticardiolipin (aCL) were quantified using enzyme-linked immunosorbent assay (ELISA) kits provided by Orgentec (Mainz, Germany). Additional autoantibodies, including anti-Sm/RNP, anti-SSA (Ro), and anti-SSB (La), were assessed via ELISA using reagents from Hycor Biomedical (Garden Grove, CA, USA). Complement components C3 and C4 were analyzed using nephelometric techniques (Siemens AG, Munich, Germany), while the functional activity of the classical complement pathway (CH50) was evaluated with an in-house hemolytic assay. All assays were conducted according to the respective manufacturers’ instructions. Laboratory analyses were performed at the Department of Laboratory Medicine, University of Debrecen. For ELISA-based determinations, five aliquots of serum (each approximately 300–400 μL) were stored in Eppendorf tubes at −80 °C until analysis.

### 2.3. Measurement of Erum Endocan (Endothelial Cell Specific Molecule-1, ESM1)

Serum endocan was determined using a commercially available quantitative sandwich ELISA (Elabscience Biotechnology, Houston, TX, USA). Values are expressed as pg/mL with 5.04–6.58% and 6.19–8.08% intra- and inter-assay precision, respectively.

### 2.4. Measurement of TNFα

Serum levels of TNFα were quantified using a TNFα-specific ELISA kit (R&D Systems Europe Ltd., Abington, UK). Assays were conducted following the manufacturer’s protocol. The intra-assay coefficients of variation (CV) ranged between 1.9% and 2.2%, while inter-assay CVs varied from 6.2% to 6.7%. Results were reported in picograms per milliliter (pg/mL).

### 2.5. Measurement of Interleukin-6 (IL-6)

Serum interleukin-6 (IL-6) concentrations were measured using a commercially available quantitative sandwich ELISA kit (R&D Systems, Abington, UK). Data are presented in picograms per milliliter (pg/mL), with intra-assay and inter-assay precision ranging from 1.7% to 4.4% and 2.0% to 3.7%, respectively.

### 2.6. MPO, MCP-1, MMP-1, MMP-3, MMP-7 and MMP-9 Enzyme-Linked Immunosorbent Assays

Serum concentrations of MPO, MCP-1, MMP-1, MMP-3, MMP-7, and MMP-9 were measured by commercially available sandwich ELISA kits (R&D Systems Europe Ltd., Abington, England). The intra- and inter-assay coefficients of variations were 1.5–2.6% and 8.0–10.8% (MPO); 4.7–7.8% and 4.6–6.7% (MCP-1); 5.4–6.1% and 6.7–10.4% (MMP-1); 5.7–6.4% and 7.0–8.6% (MMP-3); 3.7–5.1% and 4.1–4.6% (MMP-7); and 1.9–2.9% and 6.9–7.9% (MMP-9), respectively. Every ELISA assay was performed according to the manufacturer’s instructions.

### 2.7. Ultrasound-Based Evaluation of Endothelium-Dependent Vasodilatory Function in the Brachial Artery

Flow-mediated vasodilation of the brachial artery (FMD) assessments were conducted in a controlled experimental setting, following established protocols described in the literature. Participants refrained from eating for at least 8 h and avoided smoking, coffee, and tea for 18 h prior to the examination. Additionally, vasoactive medications were not permitted within 24 h before the measurements. Ultrasound evaluations were carried out on the right brachial artery using a 5–10 MHz linear transducer (Philips HD11XE, Philips, Tampa, FL, USA). Electrocardiogram (ECG) monitoring was maintained throughout the procedure. Longitudinal cross-sectional images were obtained approximately 4–7 cm above the antecubital fossa. Arterial diameter was measured at a minimum of five distinct locations. All measurements were ECG-gated and synchronized with the R-wave [20,21,22]. A blood pressure cuff was placed around the forearm and inflated to 50 mmHg above systolic pressure for 4.5 min. Upon rapid deflation of the cuff, reactive hyperemia was induced. The resulting changes in arterial diameter were recorded 60 s after cuff release. The average FMD value was derived from three consecutive measurements and expressed as a percentage, representing the relative increase in arterial diameter compared to the baseline resting state.

### 2.8. Determination of Carotid Intima-Media Ratio (cIMT)

Carotid intima-media ratio (cIMT) was assessed using duplex ultrasonography with a 5–10 MHz linear probe (Philips HD-11XE, Philips, Tampa, FL, USA). Both the left and right carotid arteries were evaluated. Prior to measurement, the common, internal, and external carotid segments were inspected for the presence of atherosclerotic plaques. In the absence of detectable plaques, cIMT measurements were carried out approximately 1 cm below the carotid bifurcation. The thickness was defined as the distance between two distinct echogenic layers: the lumen–intima interface and the media–adventitia border. A total of ten measurements were taken on each side, and the average value per artery was determined, followed by calculation of the overall mean cIMT. The final results were expressed in centimeters (cm).

### 2.9. Analysis of Arterial Stiffness Parameters

The assessment was carried out under standardized conditions following a 10-min rest period. Pulse wave velocity (PWV) and augmentation index (Aix) were evaluated using an Arteriograph device (TensioMed Kft., Budapest, Hungary). This technique relies on the concept that cardiac systole generates a pulse wave in the aorta, which is partially reflected at the aortic bifurcation. Consequently, a secondary (reflected) wave appears during late systole. The characteristics of this reflected wave are influenced by arterial stiffness—particularly of the common carotid artery—and peripheral vascular resistance, which together affect its amplitude. Reflection time (RT S35) is defined as the time interval between the primary systolic wave and the reflected systolic wave, measured when the brachial artery cuff is inflated to a pressure at least 35 mmHg above systolic levels [23,24,25]. The Aix is determined by calculating the difference between the initial and reflected systolic peaks, divided by the second (reflected) peak pressure. To estimate PWV, the anatomical distance between the aortic root and bifurcation was approximated by measuring the length from the jugular notch to the pubic symphysis. This distance was then divided by RT S35, yielding the PWV value in meters per second (m/s).

### 2.10. Statistical Analysis

Statistical evaluation was conducted using Statistica version 13.5.0.17 (TIBCO Software Inc., Palo Alto, CA, USA) and GraphPad Prism version 6.01 (GraphPad Software Inc., San Diego, CA, USA). The distribution of variables was assessed using the Kolmogorov–Smirnov test. For comparisons of continuous data, either the unpaired Student’s *t*-test or the Mann–Whitney U-test was applied, depending on the distribution characteristics. Categorical variables were analyzed using the Chi-square test. Data are reported as mean ± standard deviation for normally distributed variables or as median with interquartile range (25–75th percentile) for non-normally distributed variables. Correlations between continuous variables were evaluated using Pearson’s correlation coefficient. To identify independent predictors of accelerated atherosclerosis, a multiple linear regression analysis was performed using a backward stepwise elimination approach. A threshold of 0.05 or lower for the *p*-value was considered statistically significant.

## 3. Results

Clinical characteristics, vascular diagnostic tests, laboratory parameters, and SLE-related clinical data are summarized in Table 1. The category “DMARDs” refers to azathioprine, cyclosporine, and methotrexate therapy administered at the time of the study.

Compared to inactive SLE patients (SLE DAI < 6, n = 28), active patients (SLE DAI ≥ 6, n = 19) had significantly more antiphospholipid (aPL) antibodies. Significantly lower serum complement-4 (C4) and myeloperoxidase (MPO), as well as significantly higher monocyte chemoattractant protein-1 (MCP-1), was detected in active patients compared to the inactive group. Comparisons of cIMT, FMD, and pulse wave velocity (PWV) revealed no significant distinctions between active and inactive participants. In accordance with previous literature, serum endocan was higher in active SLE patients compared to the inactive group, however, in our study, the difference was not statistically significant (Figure 1, Table 1). We did not observe significant differences in any other serum inflammatory markers between active and inactive patients (Table 1).

A significant negative correlation was found between 50% hemolytic complement (CH50) and serum endocan in the whole patient population and in the active SLE group. Serum endocan negatively correlated with PWV and with serum complement-3 (C3) in the active SLE group (Figure 2).

Serum TNFα positively correlated with PWV in the whole study population and in inactive patients. Significant positive correlations were found between matrix metalloprotease-7 (MMP7) and TNFα in the whole study population, between C4 and TNFα in active patients, and between MCP-1 and TNFα in all study groups, while significant negative correlations were detected between CH50 and TNFα in all study groups (Figure 3).

There were significant positive correlations between PWV and serum hsCRP and between serum MCP-1 and hsCRP levels in all study groups (Figure 4).

Figure 5 presents the outcomes of univariate analyses assessing the relationship between PWV and clinical parameters in the full SLE cohort. PWV positively correlated with age, apolipoprotein B100, total cholesterol, triglyceride, hsCRP, and TNFα in the whole study population (Figure 5a). In patients with active SLE, PWV positively correlated with age, triglyceride, and MPO, but negatively correlated with endocan levels (Figure 5b). PWV positively correlated with triglyceride, apolipoprotein B100, hsCRP, and TNFα in the inactive patient group (Figure 5c).

Backward stepwise multiple regression analyses showed that hsCRP was an independent predictor of PWV in the whole patient population (β = 0.568; *p* < 0.001) and in patients with inactive SLE (β = 0.605; *p* = 0.001), respectively. In the active SLE group, serum amyloid A was found as significant predictor of PWV (β = 0.626; *p* < 0.001). Models included age, serum amyloid A, apolipoprotein B100, total cholesterol, triglyceride, hsCRP, MCP-1, and TNFα

## 4. Discussion

This study is the first to evaluate the relationship between serum endocan levels and three vascular diagnostic measures—arterial stiffness, cIMT, and FMD—in patients with SLE. Endocan has recently gained recognition as a valuable biomarker with significant diagnostic and prognostic potential in diverse clinical conditions, including COVID-19, chronic kidney disease, obstructive sleep apnea, diabetes mellitus, coronary artery disease, and preeclampsia [26]. Higher serum endocan was detected in our active SLE patients compared to the inactive group, although the difference observed did not reach statistical significance. Before our study, we hypothesized that higher serum endocan may be associated with early vascular damage, however, none of the performed vascular diagnostic test results correlated with serum endocan. A previous study reported a positive correlation between serum endocan and cIMT, however, the enrolled female SLE patients were older (by approximately 7 years), and the patients had higher BMI. Furthermore, disease activity was not considered [18]. Serum endocan values of middle-aged patients with rheumatoid arthritis were significantly higher than those of age-matched controls, and endocan showed a significantly positive correlation with disease activity score and cIMT in another study, however, younger patents were not enrolled [27]. Moreover, a previous study demonstrated significant correlation between cIMT and endocan in obese children compared to children with normal body weight [28]. Therefore, the measurement of serum endocan can help predict long-term subclinical atherosclerosis in middle-aged and elderly patients with SLE but may not be useful in young patients with a shorter disease duration, even in case of a currently active disease. Furthermore, being overweight or obese may further enhance atherogenesis in SLE and may modify the predictive role of serum endocan. Unfortunately, in our current study we enrolled only young and normal weight patients. Therefore, for future studies, the inclusion of patient populations differing in disease activity, age, and body weight is warranted to clarify in which subpopulations the measurement of endocan levels may be most informative.

Significant inverse correlations were observed between serum endocan levels and CH50 in both the entire cohort and the subgroup with active disease, alongside a negative correlation between serum endocan and complement component C3 within the active patient group. In cases of SLE, hypocomplementemia can involve decreased plasma C3, C4, or both. CH50 measures the hemolytic activity of plasma against sensitized sheep erythrocytes. This parameter reflects the capacity to generate membrane attack complexes through the classical complement pathway, thereby providing an overall assessment of complement activity spanning components C1 through C9. In patients with active SLE, CH50 levels are reduced due to enhanced consumption associated with upregulated activation of the classical complement pathway [29]. Since serum complement factors like C3 and C4 as well as the derived functional parameter CH50 are widely accepted for monitoring disease activity in patients with SLE [29,30], our results indicate a correlation between disease activity and serum endocan.

TNFα is a key proinflammatory cytokine involved in the development of atherosclerosis. Notably, TNFα was among the earliest identified cytokines associated with the pathogenesis of atherosclerotic cardiovascular disease [31]. TNFα acts as an initial mediator in the acute-phase response, contributing to the synthesis of chemokines, IL-6, and C-reactive protein (CRP), while also facilitating leukocyte recruitment during inflammation. Additionally, TNFα promotes smooth muscle cell proliferation and enhances leukocyte adhesion to endothelial cells by upregulating the expression of adhesion molecules [32]. CRP is a pentraxin primarily produced by the liver in response to proinflammatory cytokines, notably interleukin-1, IL-6, and TNFα. Circulating CRP is increased in many inflammatory conditions and even a moderate elevation is associated with a higher risk of cardiovascular disease [33]. In a Japanese cohort, maximum cIMT demonstrated a positive association with hsCRP following adjustment for age, serum E-selectin levels, and additional cardiovascular risk factors [34]. PWV was significantly related to plasma hsCRP, TNFα, and IL-6 in untreated patients with essential hypertension [35]. Another previous study observed positive correlations between TNFα, hsCRP, IL-6, IL-1β, and PWV in hypertensive patients [36]. However, to date, associations between hsCRP, TNFα, IL-6, and PWV have not been studied. In our patients, serum TNFα and hsCRP tightly correlated with PWV, however, we did not observe significant associations between inflammatory markers and vascular diagnostic parameters. A backward stepwise multiple regression analyses demonstrated hsCRP to be the best predictor of PWV in the entire and active SLE population.

Subclinical atherosclerosis, indicated by altered arterial stiffness measurements, can be identified at an early stage in SLE and significantly contributes to cardiovascular risk and associated morbidity. Current evidence highlights the pivotal role of inflammation in driving the accelerated progression of atherosclerosis in SLE patients. Our results underscore these findings. Therefore, reducing inflammation is one of the essential targets not only to slow the progression of the underlying disease, but also in decreasing CVD risk in SLE patients. It is very important to reduce disease activity and achieve remission or low disease activity as well as to administer the lowest possible dose of glucocorticoids or stop it to minimize metabolic and other side effects. The antimalarials-chloroquine and hydroxychloroquine-have been shown to be cardioprotective in SLE and reduce the production of serum inflammatory cytokines [37]. While statins have established anti-inflammatory properties in the general population, the evidence regarding their efficacy and benefit in patients with SLE remains inconclusive [38]. Recent advances in targeted therapies such as belimumab and anifrolumab have expanded treatment options for patients with SLE. Numerous additional therapeutic candidates are currently being evaluated, such as obinutuzumab, dapirolizumab pegol, and telitacicept, each targeting different mechanisms involved in SLE pathogenesis. Preliminary clinical data on chimeric antigen receptor T-cell (CART) therapy and T-cell engaging therapies indicate potential efficacy in managing severe, treatment-refractory SLE cases [39]. However, their potential anti-inflammatory and atheroprotective effects need to be investigated. Overall, future therapeutic strategies should prioritize pharmacological interventions that have been proven to reduce the degree of inflammation. During patient follow-up, the assessment of arterial stiffness parameters will likely aid in cardiovascular risk estimation. Given that disease activity is associated with enhanced inflammatory processes—which correlate with endocan levels—the measurement of serum endocan, alongside routine laboratory parameters, may also contribute to improved risk assessment in the future.

The strength of the study is that the enrolled SLE patients were young, and the diseases activity varied in a wide range. Furthermore, three widely used standardized methods were applied to evaluate early vascular damage and several soluble atherosclerosis biomarkers were assessed. Nonetheless, certain limitations of this study warrant consideration. The modest sample size of SLE patients may have limited the statistical power of our analyses. Moreover, the low ratio of male patients (a universal characteristic of SLE) does not allow the investigation of gendered differences. Further, large-scale, long-term, multicenter studies are needed to clarify the value of endocan measurement in various subpopulations of SLE patients. Assessment of more endothelium biomarkers or mechanistic research could be performed to further understand endocan’s involvement in vascular inflammation and atherogenesis. It could be considered during the evaluation of the effects of biologic DMARDs, which have been increasingly used in recent years—including agents such as belimumab, anifrolumab, and other emerging therapeutic options—on serum endocan and other vascular biomarkers. In addition, the inclusion of older healthy individuals in future studies may also contribute to a better understanding of the role of endocan. Nonetheless, our findings underscore the value of novel and conventional inflammatory markers in cardiovascular risk prediction and the importance of routine measurement of arterial stiffness parameters in young SLE patients.

## 5. Conclusions

Our findings suggest that serum endocan is associated with disease activity; however, further clinical studies involving larger patient cohorts are required to better elucidate the role of serum endocan and other glycocalyx-related biomarkers in cardiovascular risk assessment among SLE patients. Measurement of serum endocan levels, alongside conventional inflammatory markers such as hsCRP and TNFα, as well as the routine assessment of arterial stiffness parameters, should be integrated into the comprehensive management plans of young patients with SLE.

## Figures and Tables

**Figure 1 jcm-14-05955-f001:**
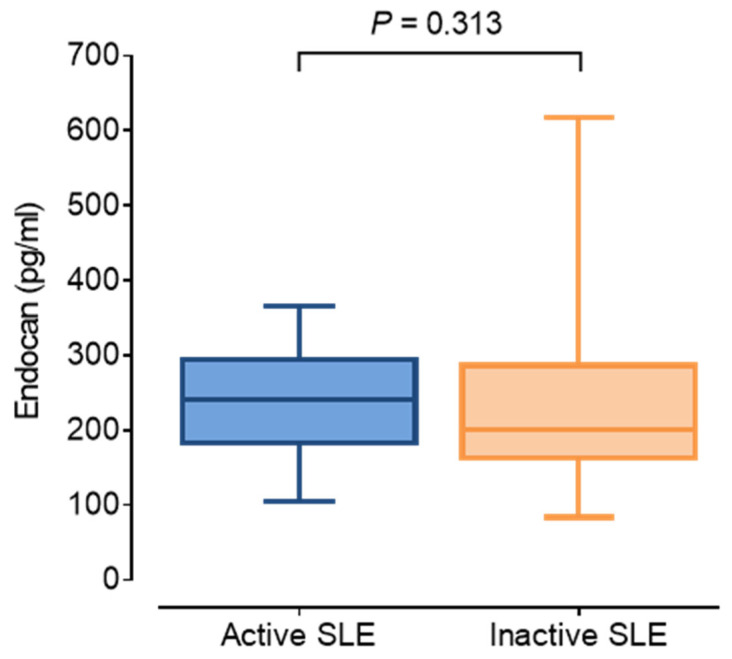
Serum endocan concentrations in patients with active (blue; SLE DAI ≥ 6) and inactive (orange; SLE DAI < 6) systemic lupus erythematosus. Abbreviations: SLE, systemic lupus erythematosus; DAI, disease activity index. Boxes indicate interquartile ranges and whiskers represent minimum and maximum values. The solid line represents median within the box.

**Figure 2 jcm-14-05955-f002:**
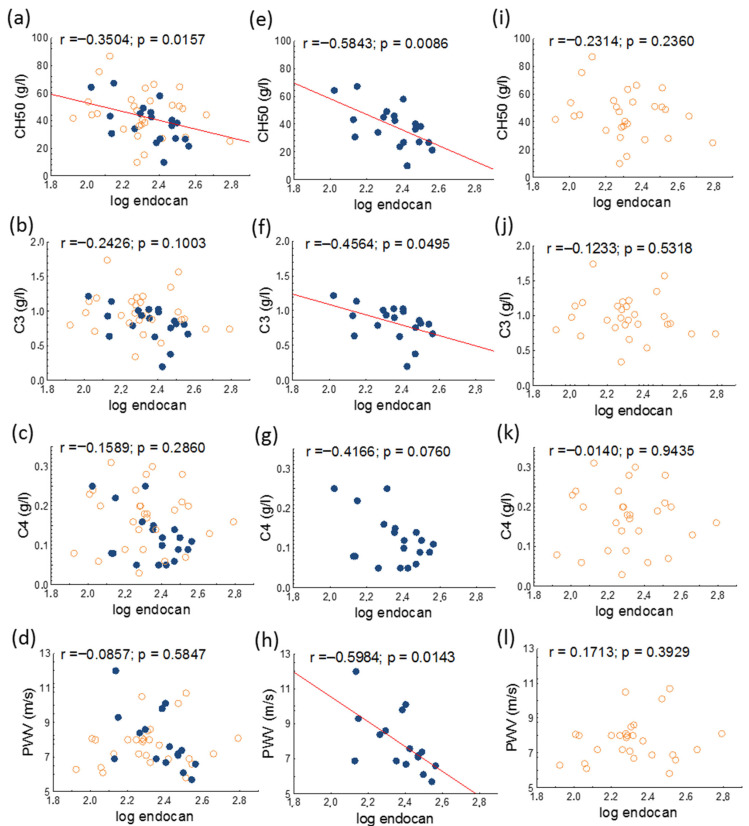
Correlations of serum endocan and CH50, C3, C4, and PWV in overall (**a**–**d**), active (represented as blue dots; (**e**–**h**)) and inactive (represented as orange circles; (**i**–**l**)) patients with SLE. Abbreviations: C3, complement 3; C4, complement 4; CH50, 50% hemolytic complement; PWV, pulse wave velocity; SLE, systemic lupus erythematosus.

**Figure 3 jcm-14-05955-f003:**
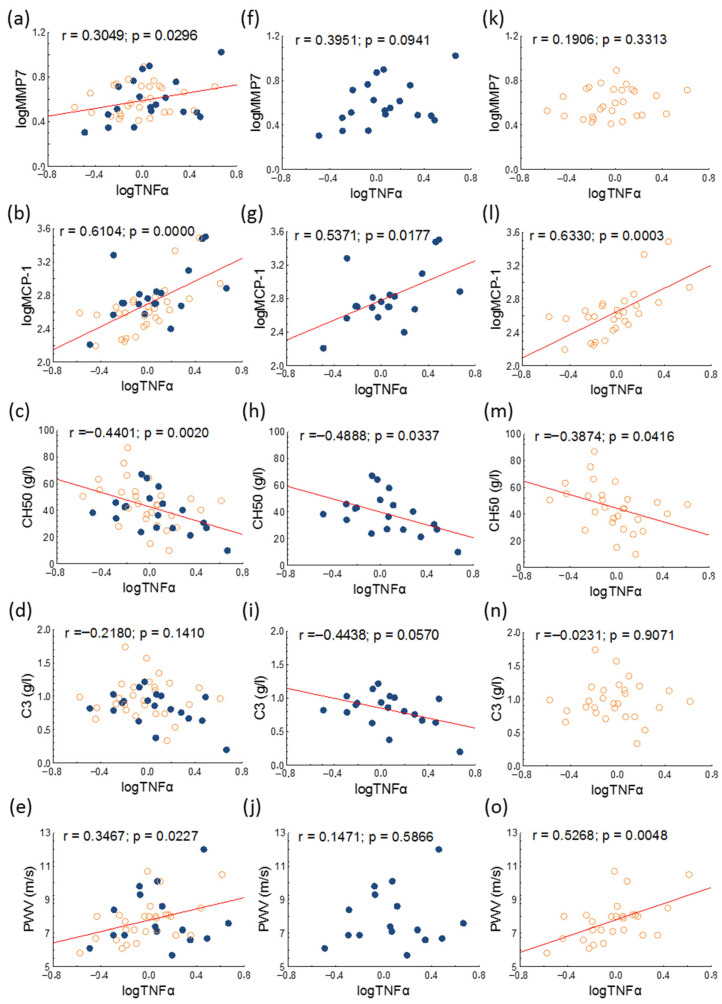
Correlations of serum TNFα and MMP7, MCP-1, CH50, C3, and PWV in overall (**a**–**e**), active (represented as blue dots; (**f**–**j**)) and inactive (represented as orange circles; (**k**–**o**)) patients with SLE. Abbreviations: C3, complement 3; CH50, 50% hemolytic complement; MCP-1, monocyte chemoattractant protein-1; MMP7, matrix metalloprotease-7; PWV, pulse wave velocity; SLE, systemic lupus erythematosus; TNFα, tumor necrosis factor-alpha.

**Figure 4 jcm-14-05955-f004:**
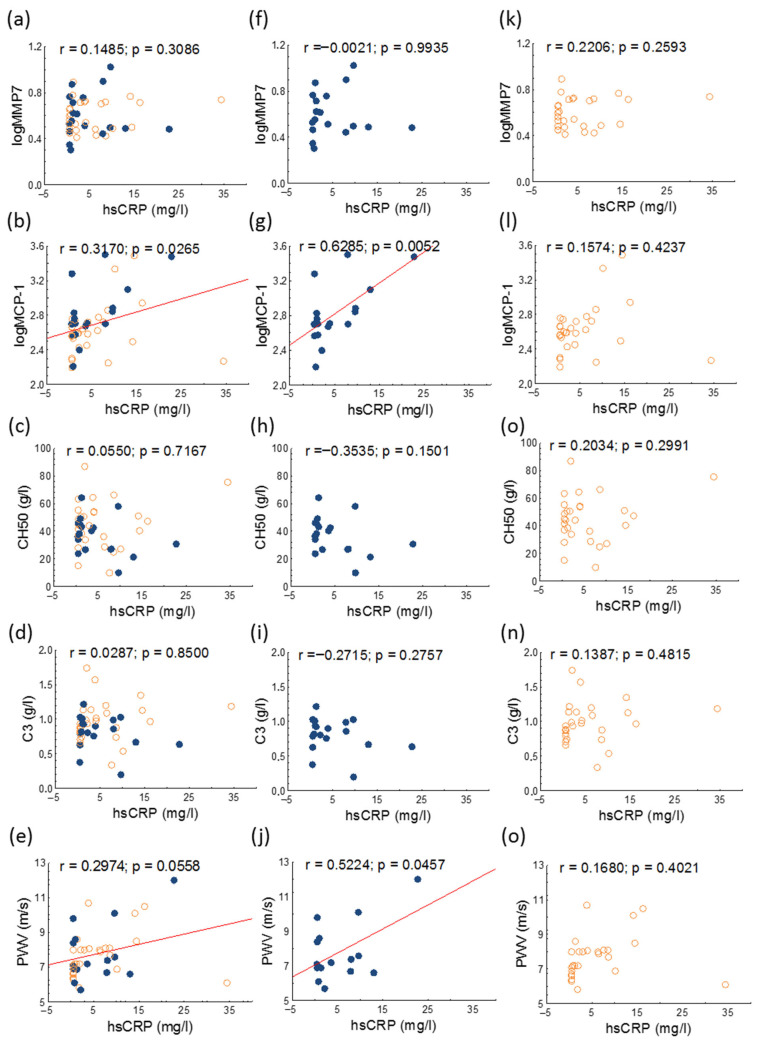
Correlations of hsCRP and MMP7, MCP-1, CH50, C3, and PWV in overall (**a**–**e**), active (represented as blue dots; (**f**–**j**)) and inactive (represented as orange circles; (**k**–**o**)) patients with SLE. Abbreviations: hsCRP, high sensitivity C-reactive protein; C3, complement 3; CH50, total complement activity; MCP-1, monocyte chemoattractant protein-1; MMP7, matrix metalloprotease-7; PWV, pulse wave velocity; SLE, systemic lupus erythematosus.

**Figure 5 jcm-14-05955-f005:**
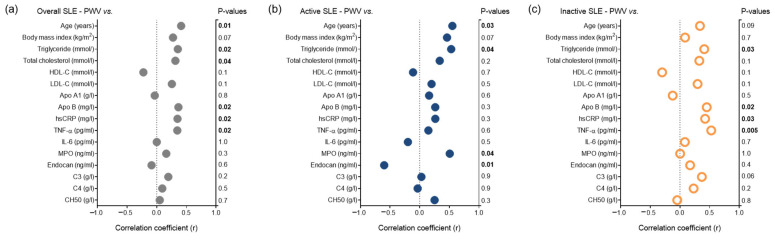
Associations of key anthropometric and laboratory variables with PWV in (**a**) the total cohort, (**b**) active SLE patients, and (**c**) inactive SLE patients. Abbreviations: Apo A1, apolipoprotein AI; Apo B, apolipoprotein B; C3, complement 3; C4, complement 4; CH50, total complement activity; HDL-C, high-density lipoprotein cholesterol; hsCRP, high sensitive C-reactive protein; IL-6, interleukin-6; MPO, myeloperoxidase; PWV, pulse wave velocity; TNF-α, tumor necrosis factor-alpha.

**Table 1 jcm-14-05955-t001:** Clinical characteristics, medication, vascular diagnostic tests, and laboratory data of study individuals.

	All Patients	SLE DAI ≥ 6	SLE DAI < 6	*p*-Value
Number of patients (n)	47	19	28	
Female (n; %)	40 (85.1)	16 (84.2)	24 (85.7)	0.845
Age (years)	31.9 ± 6.0	30.0 ± 7.2	33.3 ± 4.8	0.070
BMI (kg/m^2^)	24.4 ± 4.3	24.6 ± 5.0	24.3 ± 3.8	0.818
Smokers (n, %)	18 (38.3)	7 (36.8)	11 (39.3)	0.771
aPL antibodies (n, %)	7 (14.9)	5 (26.3)	2 (7.1)	**<0.001**
Prednisolone (n, %)	47 (100.0)	19 (100.0)	28 (100.0)	1.000
Chloroquine (n, %)	22 (46.8)	8 (42.1)	14 (50.0)	0.256
NSAIDs (n, %)	17 (36.2)	7 (36.8)	10 (35.7)	0.883
DMARDs (n, %)	32 (68.1)	14 (73.7)	18 (64.3)	0.126
SLE DAI	4.0 (2.0–10.0)	10.0 (8.0–12.0)	4.0 (2.0–4.0)	**<0.001**
Vascular diagnostic tests				
cIMT (cm)	0.049 ± 0.010	0.052 ± 0.012	0.047 ± 0.008	0.156
FMD (%)	7.8 (5.8–12.0)	7.4 (5.6–11.1)	8.1 (6.6–12.3)	0.502
Aix (%)	−15.4 (−36.8–11.9)	−23.2 (−37.6–8.0)	−10.8 (−35.9–15.2)	0.386
PWV (m/s)	7.8 ± 1.4	7.9 ± 1.7	7.7 ± 1.2	0.704
Laboratory parameters				
Complement 3 (g/L)	0.92 ± 0.28	0.83 ± 0.25	0.98 ± 0.29	0.068
Complement 4 (g/L)	0.15 ± 0.08	0.12 ± 0.06	0.17 ± 0.08	**0.021**
CH50 (g/L)	42.49 ± 16.39	38.54 ± 14.74	45.2 ± 17.2	0.176
Endocan (pg/mL)	208.6 (175.5–294.5)	241.4 (183.2–294.5)	200.3 (167.0–277.9)	0.313
hsCRP (mg/L)	2.1 (0.57–8.00)	1.7 (0.76–8.00)	2.5 (0.54–8.04)	0.822
Interleukin-6 (pg/mL)	0.54 (0.0–3.25)	0.85 (0.0–2.51)	0.50 (0.07–3.44)	0.845
TNFα (pg/mL)	0.99 (0.63–1.46)	1.13 (0.62–1.91)	0.96 (0.64–1.32)	0.398
Myeloperoxidase (ng/mL)	465.8 (301.9–835.0)	385.8 (276.8–537.2)	648.3 (351.9–957.0)	**0.046**
Serum amyloid A (μg/mL)	4.08 (1.49–14.07)	3.76 (2.53–14.07)	7.49 (1.40–14.33)	0.756
MCP-1 (pg/mL)	473.2 (362.4–649.9)	512.8 (473.2–772.3)	395.5 (299.0–545.2)	**0.028**
MMP3 (ng/mL)	31.5 (15.0–50.8)	29.2 (16.2–45.9)	32.1 (13.5–53.1)	0.905
MMP7 (ng/mL)	3.58 (3.04–5.26)	3.39 (2.93–5.75)	3.82 (3.05–5.23)	0.770
MMP9 (ng/mL)	298.4 (176.3–479.9)	215.0 (98.9–324.8)	339.6 (246.8–687.5)	**0.011**
Triglyceride (mmol/L)	1.30 (1.00–1.90)	1.25 (1.00–1.90)	1.35 (1.90–1.15)	0.893
Total cholesterol (mmol/L)	4.5 ± 1.1	4.3 ± 1.1	4.6 ± 1.1	0.440
HDL-C (mmol/L)	1.2 ± 0.4	1.2 ± 0.3	1.3 ± 0.5	0.441
LDL-C (mmol/L)	2.6 ± 0.8	2.6 ± 0.9	2.5 ± 0.8	0.554
Apolipoprotein B100 (g/L)	0.85 ± 0.27	0.88 ± 0.28	0.84 ± 0.26	0.618
Apolipoprotein A1 (g/L)	1.38 ± 0.33	1.29 ± 0.26	1.44 ± 0.37	0.138

Data are presented as means ± standard deviation or medians (lower-upper quartiles). The *p*-values denote the statistical difference between active (SLE DAI ≥ 6) and inactive (SLE DAI < 6) SLE patients. Continuous variables were calculated by un-paired *t*-test or Mann–Whitney U-test. Categorical variables were calculated by Chi-square test. Significant *p*-values are bolded. Abbreviations: Aix: augmentation index; aPL: antiphospholipid antibodies; APS: anti-phospholipid syndrome; BMI: body mass index; CH50: 50% hemolytic complement; cIMT: carotid intima-media thickness; HDL-C: high-density lipoprotein cholesterol; hsCRP: high sensitivity C-reactive protein; DMARDs: disease-modifying anti-rheumatic drugs; FMD: flow-mediated dilation; LDL-C: low-density lipoprotein cholesterol; MCP-1: monocyte chemoattractant protein-1; MMP3, 7, 9: matrix metalloprotease3, 7, 9; NSAIDs: nonsteroidal anti-inflammatory drugs; PWV: pulse wave velocity; SLE: systemic lupus erythematosus; SLE DAI: systemic lupus erythematosus disease activity index; TNFα: tumor necrosis factor-alpha.

## Data Availability

All data generated or analyzed during this study are included in this published article. All data generated or analyzed during the current study are available from the corresponding author on reasonable request.

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
