# Peer review of "Assessment of Serum Endocan Levels and Their Associations with Arterial Stiffness Parameters in Young Patients with Systemic Lupus Erythematosus"

_jcm, 2025, doi:10.3390/jcm14175955_

Round 1

Reviewer 1 Report

Comments and Suggestions for Authors

Assessment of serum endocan levels and their associations with arterial stiffness parameters in young patients with systemic lupus erythematosus

This manuscript covers an interesting topic .However ,some points need to be addressed

Abstract

  • The existing abstract gives some background, but it should start more directly with a clear and brief statement that tells readers what the study was trying to do.
  • The abstract might make it clearer and more evident which trends are statistically significant and which are not by putting greater emphasis on the most relevant ones.
  • At the end of the abstract, write a clear phrase that sums up the study's broader implications or next actions in a few words.

Introduction

  • The first phrase talks about cardiovascular disease (CVD) as a common cause of death around the world that is more common in autoimmune illnesses like SLE. This is too general and might be more interesting and focused on your study issue.
  • The introduction goes from talking about the broader problem of not taking CVD risk in SLE seriously enough to talking about traditional and non-traditional risk factors, and then to the glycocalyx and endocan. All of the ideas are important, but the transitions could be smoother if they were more clearly linked.
  • It's fantastic that you say that the link between endocan and arterial stiffness and FMD in SLE hasn't been examined yet. Make this gap more noticeable to make the study question stand out.
  • For the sake of brevity, phrases like "the whole group of diseases" or "the clinical significance and prognostic role of novel vascular, metabolic, and inflammatory biomarkers in SLE are under continuous investigation" could be cut down or left out.

Methods

  • Your Methods section is well-organized into subsections, but make sure that all of the names and formats are the same so that it is easy to find your way around. For example, "2.3 Serum Endocan Measurement" may be changed to "2.3 Measurement of Serum Endocan" to match other parts like 2.4 "TNFα measurement."
  • To minimize confusion, make a clear list of all the criteria for inclusion and exclusion (for example, don't include people with serious comorbidities or infections).
  • Say whether and how the sample size was chosen, and if the study was strong enough to find certain differences or correlations.
  • Say which tests you ran for data that was normally distributed and which tests you used for data that wasn't.
  • You give the names of the applications used for statistical analysis , but make sure that the version numbers are always shown.
  • You include approval numbers and consent, but you may also add the dates when the study took place to give it some perspective in time.
  • Results
  • You talk about comparing active and inactive SLE patients, but you don't always say which outcomes go with which group. To avoid confusion, always connect results to groups of patients.

    Say things like "In patients with active SLE (SLEDAI ≥6)..." or "Compared to patients who aren't active..."
  • If the data is regularly distributed, report it as mean ± SD. If it is skewed, report it as median with interquartile range.
  • Avoid repetitive phrases or unnecessarily restating results that are already summarized in tables and figures.

Discussion

  • Instead of just summarizing previous studies, use them to back up your claims and show where your work fills in gaps. For instance, look at biomarker levels and correlations while taking into account the patient's age, sex, and disease activity.
  • Talk about how your results help with risk stratification or managing cardiovascular risk in SLE. Talk about how endocan might be used in place of traditional biomarkers and arterial stiffness measurements, stressing how important it is to regulate inflammation.
  • Suggest specific future steps, such as larger groups, longer-term trials, more endothelium biomarkers, or mechanistic research to further understand endocan's involvement.
  • Avoid saying the same thing over and over again (for example, saying that there is a positive correlation in different situations).

    Don't use tentative language when your research backs up a conclusion, but be careful and fair.

Author Response

Reviewer#1

Comments and Suggestions for Authors 

This manuscript covers an interesting topic .However ,some points need to be addressed

Response: First, thank you for your positive opinion about our paper. Please find our answers to the questions and comments as follows.

Abstract

  • The existing abstract gives some background, but it should start more directly with a clear and brief statement that tells readers what the study was trying to do.

Response: Thank you for the suggestion. We have revised the background section of the abstract, formulating the aim of the study in a brief and direct manner (Ln 21-25).

  • The abstract might make it clearer and more evident which trends are statistically significant and which are not by putting greater emphasis on the most relevant ones.

Response: Thank you, we revised the results section of the abstract according to your suggestions (Ln 35-43).

  • At the end of the abstract, write a clear phrase that sums up the study's broader implications or next actions in a few words.

Response: Thank you for your comment, we completed the conclusion section of the abstract (Ln 44-46)

Introduction

  • The first phrase talks about cardiovascular disease (CVD) as a common cause of death around the world that is more common in autoimmune illnesses like SLE. This is too general and might be more interesting and focused on your study issue.

Response: Thank you for the comment. We rephrased the initial part of the introduction section (Ln 51-61).

  • The introduction goes from talking about the broader problem of not taking CVD risk in SLE seriously enough to talking about traditional and non-traditional risk factors, and then to the glycocalyx and endocan. All of the ideas are important, but the transitions could be smoother if they were more clearly linked.

Response: Thank you for the comment, we tried to link these topics better (Ln 73-74).

  • It's fantastic that you say that the link between endocan and arterial stiffness and FMD in SLE hasn't been examined yet. Make this gap more noticeable to make the study question stand out.

Response: Thank you for the suggestion. We rephrased this part of the introduction to highlight the gap in this field (Ln 99-103).

  • For the sake of brevity, phrases like "the whole group of diseases" or "the clinical significance and prognostic role of novel vascular, metabolic, and inflammatory biomarkers in SLE are under continuous investigation" could be cut down or left out.

Response: Thank you for the comment, "the whole group of diseases” was deleted (ln 54-55). The other sentence was shortened (Ln 66-68).

Methods

  • Your Methods section is well-organized into subsections, but make sure that all of the names and formats are the same so that it is easy to find your way around. For example, "2.3 Serum Endocan Measurement" may be changed to "2.3 Measurement of Serum Endocan" to match other parts like 2.4 "TNFα measurement."
  • Response: Thank you for the comments. "2.3 Serum Endocan Measurement" was changed to "2.3 Measurement of Serum Endocan" (Ln 166), while 2.4 "TNFα measurement." was changed to “2.4. Measurement of TNFα”(Ln 170). Furthermore, “2.5. Interleukin-6 (IL-6) measurement” was changed to “ 2.5. Measurement of Interleukin-6 (IL-6)” (Ln 175).

  • To minimize confusion, make a clear list of all the criteria for inclusion and exclusion (for example, don't include people with serious comorbidities or infections).

Response: Thank you, we clarified the criteria for inclusion and exclusion (Ln 122-123)

  • Say whether and how the sample size was chosen, and if the study was strong enough to find certain differences or correlations.

Response: We conducted a power analysis to determine sample size with the 80% thresholds for power. Sample size was assessed by a two-sample Wilcoxon Mann-Whitney U-Test with the following serum endocan levels: 240 pg/ml (for active SLE, group 1) and 200 pg/ml (for inactive SLE, group 2). The alpha level was 0.05, with 0.8 (80%) of desired power. Required sample sizes for both groups were calculated as n1 = 16 and n2 = 16, respectively.

  • Say which tests you ran for data that was normally distributed and which tests you used for data that wasn't.

Response: The Kolmogorov–Smirnov test was used to determine the normality of the data. The Mann–Whitney u-test were performed to describe the difference between continuous variables with non-normal distribution (Ln 273-276).

  • You give the names of the applications used for statistical analysis , but make sure that the version numbers are always shown.

Response: The version numbers are checked and shown in the text (Ln 271-273).

  • You include approval numbers and consent, but you may also add the dates when the study took place to give it some perspective in time.

Response: Thank you for this comment, we added the dates when the study took place (Ln 131-132)

Results

  • You talk about comparing active and inactive SLE patients, but you don't always say which outcomes go with which group. To avoid confusion, always connect results to groups of patients.

Say things like "In patients with active SLE (SLEDAI ≥6)..." or "Compared to patients who aren't active..."

 Response: Thank you for the suggestion, we corrected these sentences (Ln 314, 322-323).

  • If the data is regularly distributed, report it as mean ± SD. If it is skewed, report it as median with interquartile range.

Response: Indeed, this is the way that we reported the results in Table 1. For example, BMI was normally distributed, therefore we reported mean ± SD. In contrast, serum endocan were not normally distributed, we reported median with interquartile range (Ln 298, Table 1).

  • Avoid repetitive phrases or unnecessarily restating results that are already summarized in tables and figures.

Response: Thank you for the comment. We tried to highlight only the most important positive findings, and rest of the data summarized in table and figures are not discussed in detail. Still, we deleted an unnecessary sentence (Ln 2331-332).

Discussion

  • Instead of just summarizing previous studies, use them to back up your claims and show where your work fills in gaps. For instance, look at biomarker levels and correlations while taking into account the patient's age, sex, and disease activity.

Response: Thank you for this comment. We rephrased this part according to your suggestions (Ln 394, 397-398, 399-400).

  • Talk about how your results help with risk stratification or managing cardiovascular risk in SLE. Talk about how endocan might be used in place of traditional biomarkers and arterial stiffness measurements, stressing how important it is to regulate inflammation.

Response: We completed the discussion section with these thoughts (Ln 462-468).

  • Suggest specific future steps, such as larger groups, longer-term trials, more endothelium biomarkers, or mechanistic research to further understand endocan's involvement.

Response: Thank you for the comment, we added these suggestions to the limitation section of the discussion (Ln 475-479).

  • Avoid saying the same thing over and over again (for example, saying that there is a positive correlation in different situations).
    Response: Thank you for the comment. We tried to do our best to eliminate repetitions throughout the manuscript.

Don't use tentative language when your research backs up a conclusion, but be careful and fair.

Response: Thank you, we rephrased the conclusion (Ln 488-491).

Considering the mentioned points, please revalue our manuscript. Hoping that your decision will be favorable, we would like to thank you for your support.

Sincerely,

Mariann Harangi, MD, PhD, DSc

Institute of Medicine

University of Debrecen Faculty of Medicine

Nagyerdei krt. 98, H-4032 Debrecen, Hungary.

Tel/Fax: + 36 5244 2101

E-mail: harangi@belklinika,  harangi.mariann@med.unideb.hu

Reviewer 2 Report

Comments and Suggestions for Authors
  • The results cannot be considered as specific to SLE because of the lack of healthy control group.
  • It is presumable levels of serum biomarkers in relation with inflammatory cytokines would be altered by immunosuppressive drugs. Despite this, details in immunosuppressive therapy such as dosage of glucocorticoids are not sufficiently provided. It is not clear what drugs are included in ‘DMARDs’ in Table 1. i.e. usage rates of anti-SLE drugs such as MMF, CY, belimumab and so on should be provided. The authors mentioned relation between endocan and TNFα/IL-6. If patients received bDMARDs against TNFα/IL-6, this would affect endocan levels irrespective of disease activity of SLE.
  • It is not clear ‘Nephritis syndrome’ is included in Supplementary Table 1 while the authors declared they excluded patients with lupus nephritis on page 3, line 103.

Author Response

Comments and Suggestions for Authors

Response: First, thank you for your positive opinion about our paper. Please find our answers to the questions and comments as follows.

  • The results cannot be considered as specific to SLE because of the lack of healthy control group.

Response: We agree with the reviewer’s comment. We can only draw conclusions based on our study population; therefore, we did not generalize the results. The investigation of a healthy control population may also be important in future studies. However, there is concern that in an age- and sex-matched healthy population (young females), the extent of vascular inflammation and atherosclerosis may be insufficient to allow for the meaningful evaluation of endocan or other vascular biomarkers. However, in the future, studies involving older healthy individuals may provide more valuable insights. We have added this suggestion to the limitations section (Ln 482-483). 

  • It is presumable levels of serum biomarkers in relation with inflammatory cytokines would be altered by immunosuppressive drugs. Despite this, details in immunosuppressive therapy such as dosage of glucocorticoids are not sufficiently provided. It is not clear what drugs are included in ‘DMARDs’ in Table 1. i.e. usage rates of anti-SLE drugs such as MMF, CY, belimumab and so on should be provided. The authors mentioned relation between endocan and TNFα/IL-6. If patients received bDMARDs against TNFα/IL-6, this would affect endocan levels irrespective of disease activity of SLE.

Response: Thank you for the comment. In the table, the category of "DMARDs" refers to azathioprine, cyclosporine, and methotrexate therapy administered at the time of the study, in accordance with our national guidelines. This information has been added to the Results section (Ln 296-297). At the time of the study, biological DMARDs were not available in our country, thus their potential influence can be excluded. However, their impact may indeed be relevant in future studies, and we have added this consideration to the Limitations section (479-482).

  • It is not clear ‘Nephritis syndrome’ is included in Supplementary Table 1 while the authors declared they excluded patients with lupus nephritis on page 3, line 103.

Response: Thank you for the comment. Only patients with active nephritis were excluded from the study (Ln 122). However, there were three patients with positive history of nephritis (Supplementary table 1).

Considering the mentioned points, please revalue our manuscript. Hoping that your decision will be favorable, we would like to thank you for your support.

Sincerely,

Mariann Harangi, MD, PhD, DSc

Institute of Medicine

University of Debrecen Faculty of Medicine

Nagyerdei krt. 98, H-4032 Debrecen, Hungary.

Tel/Fax: + 36 5244 2101

E-mail: harangi@belklinika,  harangi.mariann@med.unideb.hu

Round 2

Reviewer 1 Report

Comments and Suggestions for Authors

The manuscript improved however ,the similarity index is very high 

Author Response

Thank you for your valuable and thorough review, which has significantly contributed to improving the quality of the manuscript.
The high similarity index was primarily due to an overlap in the methodological description with one of our previous publications (22%). At the Editor's request, we revised this section as much as possible. As a result, the similarity index of the revised (R1) manuscript—excluding the acknowledgements and keywords—is now significantly lower. According to Turnitin, which we use, it is currently only 35%, with an 11% overlap with the previous article (pdf attached). The overlap with all other sources is between 1% and 4%.

I hope this is acceptable to both you and the journal.
Thank you once again for your assistance with the manuscript.

Reviewer 2 Report

Comments and Suggestions for Authors

(No further comments)

Author Response

Thank you for your valuable and thorough review, which has significantly contributed to improving the quality of the manuscript.